# MECHANISM OF CLEAN-PRIORITY LEARNING IN EARLY STOPPED NEURAL NETWORKS OF INFINITE WIDTH

## ABSTRACT

When random label noise is added to a training dataset, the prediction error of a neural network on a label-noise-free test dataset initially improves during early training but eventually deteriorates, following a U-shaped dependence on training time. This behaviour is believed to be a result of neural networks learning the pattern of clean data first and fitting the noise later, a phenomenon that we refer to as *clean-priority learning*. In this study, we aim to explore the learning dynamics underlying this phenomenon. We demonstrate that, in the early stage of training, the update direction of gradient descent is determined by the clean samples of training data, leaving the noisy samples have minimal to no impact, resulting in a prioritization of clean learning. Moreover, we show both theoretically and experimentally, as the clean-priority learning goes on, the dominance of the gradients of clean samples over those of noisy samples diminishes, and finally results in a termination of the clean-priority learning and fitting of the noisy samples.

## 1 INTRODUCTION

Early stopping is an popular practice to achieve good performance in machine learning. The effectiveness of early stopping is evident in the setting where random label noise is injected to the training dataset while the test dataset remains intact. This noisy-training-label setting is common in the literature to investigate properties of neural networks (see for example (Zhang et al., 2021) and (Nakkiran et al., 2021)). In this setting, The test prediction error often exhibits a U-shaped dependence on training time, with an initial decrease followed by an increase after the early stopping point, see Figure 1. Interestingly, in the intermediate steps, especially around the early stopping point, the test performance can be significantly better than the label noise level added to the training set (below the dashed line).

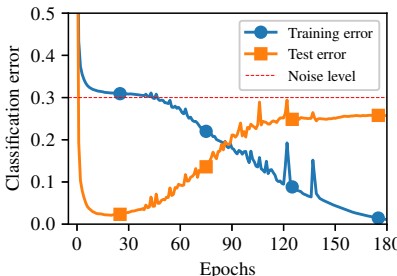

Figure 1: Classification on MNIST using CNN. Test error exhibits a U-shaped curve, and can be significantly lower than the noise level during training.

Prior studies (Arpit et al., 2017; Li et al., 2020) interpret this U-shaped behaviour as a result of neural network (NN) first learning the patterns in the clean data and overfitting the noise at a later stage. We coin the term *clean-priority learning* to describe this phenomenon. Although this is intuitively correct, the underlying mechanism of *clean-priority learning* phenomena remains unclear. Specifically, *how can the model tell clean samples from noisy ones (i.e., those with corrupted labels), without access to the ground-truth label? Furthermore, if it somehow learned the clean samples in the early stage, why the model performance deteriorate later on?* In this paper we address the above puzzling and fundamental questions, for infinitely wide neural networks.

At the outset, we analyze the configuration of sample-wise gradients on the training dataset, at initialization of the infinitely wide neural network. Our analysis reveals that samples within the same class (before label corruption), which are presumably more similar to each other, tend to have their sample-wise gradients relatively closer in vector directions (compared to the samples from different classes). The label corruption, which flips the label to a different class, flips the corresponding sample-wise gradient to its opposite direction. Consequently, the sum of the noisy sample gradients

is in sharp opposite direction of that of the clean sample gradients. It is worth noting that, due to the "transition to linearity"(Liu et al., 2020) of infinitely wide neural network, these sample-wise gradients do not change direction during training.

The key observation is that, due to the dominance of the population of the clean samples, in the early stage of learning, the gradient of noisy segment is cancelled out, and essentially makes no contribution on the gradient descent (GD) update direction[1]. It is also worth noting that almost all clean sample-wise gradient vectors "agree" with the GD update (i.e., have positive projection), while almost all noisy sample-wise gradients are "against" the GD update. As a result, the individual loss on each clean sample is decreased, and that on each noisy sample is increased. Hence, we see that, in the early stage, the GD algorithm is determined by the clean samples and exhibits the clean-priority learning.

We further show that as the clean-priority learning process continues, the clean segment gradient's dominance in magnitude over the noisy segment gradually diminishes while keeping opposite in direction, due to the decrease in the magnitude of clean segment gradients and the increase in the magnitude of noisy segment gradients. This is particularly evident around the early stopping point, where the dominance almost vanishes and the noisy segment gradient begins to make a meaningful contribution, causing the model start to fit the noisy samples, which is expected to hurt the model's performance.

We experimentally verify these findings on neural networks with finite but large widths. These experimental results also suggest that our findings may extend to finite width scenarios.

In summary, we make the following contributions:

- In the early stage of learning of neural networks, the noisy samples contribution to the GD update is cancelled out by that of the clean samples, which is the key mechanism underlying clean-priority learning.

- The clean-priority learning behavior gradually fades as the dominance of the clean segment diminishes, particularly around the early stopping point. We experimentally verify our findings on deep neural networks on various classification problems.

- For fully connected networks with mild assumption on data we theoretically prove our empirical observation.

The paper is organized as follows: in Section 2, we describe the setup of the problems and introduce necessary concepts and notations. In Section 3, we analyze the sample-wise gradients at initialization of neural network, for binary classification. In Section 4, we show the learning dynamics, especially the clean-priority learning, on binary classification. In Section 5, we extend our study and findings to multi-class classification problems.

## 1.1 RELATED WORKS

Early stopping is often considered as a regularization technique and is widely used in practice to obtain good performance for machine learning models (Gal & Ghahramani, 2016; Graves et al., 2013). Early stopping also received a lot theoretical analyses, both on non-neural network models, especially linear regression and kernel regression (Yao et al., 2007; Ali et al., 2019; Xu et al., 2022; Shen et al., 2022), and on neural networks (Zhang et al., 2021; Ji et al., 2021).

Prior studies have experimental observed the phenomenon of clean-priority learning: when random label noise presents, neural networks fit the clean data first and "overfit" the noise later on (Arpit et al., 2017; Bai et al., 2021; Ren et al., 2022). For example, Ren et al. (2022) experimentally investigated the learning paths of some "hard" samples, and observed that these learning paths have "zig-zag" patterns and show an ability of refining "bad" samples. Based on experimental investigation of critical samples, the work (Arpit et al., 2017) hypothesizes that clean samples have simple patterns and are learned by neural networks first. Even with these interesting experimental investigations, it remains unclear why clean samples represent simple patterns and what fundamental mechanism makes for the clean samples learned before the noisy ones. Instead, our work focuses on studying the underlying mechanism and theoretically explaining these phenomena from a fundamental point of view. Some theoretical work also shed a bit light on this topic. For example, the work (Li et al., 2020),

---

[1]To be more precise, the only effect of the noisy segment gradient is resulting in a smaller GD step size.

assuming (almost) perfectly cluster-able data and uniform conditioning on Jacobian matrices, proves that clean data are fit by two-layer neural networks in an early stage. However, this data assumption requires that, at the same location of each noisy sample, there must exist several (at least $1/\delta$, with $\delta \in (0, 1/2)$ being the label noise level) clean samples. Namely, each noisy sample must be covered by more clean samples. This assumption is often not met by actual datasets. Another work (Frei et al., 2021), in the simple setting of linearly separable data (before label corruption) and two-layer neural network, proved that a good generalization can be achieved. While these theoretical work are encouraging, the settings remain limited.

## 2 PROBLEM SETUP AND PRELIMINARY

In this paper, we consider supervised classification problems.

**Datasets.** There is a training dataset $\mathcal{D} \triangleq \{(\mathbf{x}_i, y_i)\}_{i=1}^n$ of size $|\mathcal{D}| = n$, where $\mathbf{x}_i \in \mathbb{R}^d$ is the input feature and $y_i$ is the label . For binary (2-class) classification, the label $y_i \in \{0, 1\}$ is binary; for multi-class classification, the label is one-hot encoded, $y_i \in \mathbb{R}^C$, where $C$ is the number of classes.

We assume the labels in $\mathcal{D}$ are randomly corrupted. Specifically, if denote $\hat{y}$ as the ground truth label of $(\mathbf{x}_i, y_i) \in \mathcal{D}$, there exists a non-empty set $\mathcal{D}_{noise} \triangleq \{(\mathbf{x}, y) \in \mathcal{D} : y \neq \hat{y}\}$. Furthermore, the labels $y_i$ in $\mathcal{D}_{noise}$ is uniformly randomly distributed across all the class labels except $\hat{y}_i$. We call $\mathcal{D}_{noise}$ as the *noisy segment* and its elements as *noisy samples*. We also define the *clean segment* $\mathcal{D}_{clean}$ as the compliment, i.e., $\mathcal{D}_{clean} = \mathcal{D} \backslash \mathcal{D}_{noise}$, and call its elements as *clean samples*. Denote $\hat{\mathcal{D}}$ as the ground-truth-labeled dataset: $\hat{\mathcal{D}} \triangleq \{(\mathbf{x}_i, \hat{y}_i)\}_{i=1}^n$. The *noise level* $\delta$ is defined as the ratio $|\mathcal{D}_{noise}|/|\mathcal{D}|$. In this paper, we set $\delta < 0.5$, i.e., the majority of training samples are not corrupted.

We further denote $\mathcal{D}^{(c)} \subset \mathcal{D}$, $c \in \{1, 2, \cdots, C\}$, as the set of samples with ground truth label $c$. We also define $\mathcal{D}_{clean}^{(c)} \triangleq \mathcal{D}^{(c)} \cap \mathcal{D}_{clean}$ and $\mathcal{D}_{noise}^{(c)} \triangleq \mathcal{D}^{(c)} \cap \mathcal{D}_{noise}$ as the class-specific clean/noise segment, respectively. In addition, there is a test dataset $\mathcal{D}_{test}$ which is i.i.d. drawn from the same data distribution as the training set $\mathcal{D}$, except that the labels of test set $\mathcal{D}_{test}$ are not corrupted.

**Optimization.** Given a dataset $\mathcal{S}$ and a model $f$ which is parameterized by $\mathbf{w}$ and takes an input $\mathbf{x}$, we define the loss function as

$$L(\mathbf{w}; \mathcal{S}) = \frac{1}{|\mathcal{S}|} \sum_{(\mathbf{x}_i, y_i) \in \mathcal{S}} l(\mathbf{w}; \mathbf{x}_i, y_i), \tag{1}$$

with $l(\mathbf{w}; \mathbf{x}_i, y_i) \triangleq l(f(\mathbf{w}; \mathbf{x}_i), y_i)$ is evaluated on a single sample. We use ReLU neural networks as the model, which are defined as:

$$f = \sigma_{\text{out}}(h(\mathbf{w}; \mathbf{x})), \quad h(\mathbf{w}; \mathbf{x}) = W^{(L+1)} \sigma \left( \sqrt{\frac{2}{m}} W^{(L)} \cdots \sigma \left( \sqrt{\frac{2}{m}} W^{(1)} \mathbf{x} \right) \right), \tag{2}$$

Here, $\sigma(\cdot) = \max(\cdot, 0)$ is the ReLU activation function, $\mathbf{w} = (W^{(L+1)}, W^{(L)}, \cdots, W^{(1)})$ represent the parameters. Each individual parameter is i.i.d. initialized using the normal distribution $\mathcal{N}(0, 1)$. $m$ is the network width, and we are interested in the infinite width limit $m \to \infty$.

For binary classification, the network $f$ has one output neuron, $\sigma_{\text{out}}$ is the *sigmoid* function, and $l$ is the logistic loss; for multi-class classification, $f$ has $C$ output neurons, $\sigma_{\text{out}}$ is the *softmax* function, and $l$ is the cross entropy loss. In both cases, each output neuron has a value in $(0, 1)$.

We minimize the empirical loss function $L(\mathbf{w}) \triangleq L(\mathbf{w}; \mathcal{D})$ using gradient descent (or its stochastic variants) which has the following update form:

$$\mathbf{w}_{t+1} = \mathbf{w}_t - \eta \nabla L(\mathbf{w}_t; \mathcal{D}) = \mathbf{w}_t - \eta \frac{1}{n} \sum_{(\mathbf{x}_i, y_i) \in \mathcal{D}} \nabla l(\mathbf{w}_t; \mathbf{x}_i, y_i). \tag{3}$$

Here, $\nabla l(\mathbf{w}; \mathbf{x}_i, y_i)$ is the gradient of loss $l(\mathbf{w}; \mathbf{x}_i, y_i)$ w.r.t. the neural network parameters $\mathbf{w}$.

**Sample-wise gradients.** We call $\nabla l(\mathbf{w}; \mathbf{x}_i, y_i)$ *sample-wise gradient*, as it is evaluated on a single sample, and denote it as $\nabla l_i(\mathbf{w})$ for short. As there are $n$ samples in $\mathcal{D}$, at each point $\mathbf{w}$ in the parameter space, we have $n$ sample-wise gradients.

Denote $h(\mathbf{w}; \mathbf{x})$ as the pre-activation output neuron(s), which is in $\mathbb{R}$ for binary classification, and is in $\mathbb{R}^C$ for multi-class classification. The sample-wise gradient for a given sample $(\mathbf{x}_i, y_i)$ has the following form (Bishop & Nasrabadi, 2006):

$$\nabla l_i(\mathbf{w}) = (f(\mathbf{w}; \mathbf{x}_i) - y_i)\nabla h(\mathbf{w}; \mathbf{x}_i). \tag{4}$$

Note that the above expression is a scalar-vector multiplication for binary classification, and is a vector-matrix multiplication for multi-class classification.

We further denote the collection of sample-wise gradients for the segment $\mathcal{D}_k^{(c)}$, for $k \in \{clean, \ noise\}$ and $c \in [C]$, by $\mathcal{G}_k^{(c)}(\mathbf{w})$, and define the corresponding *segment gradient* as $g_k^{(c)}(\mathbf{w}) \triangleq \sum_{\nabla l(\mathbf{w}) \in \mathcal{G}_k^{(c)}} \nabla l(\mathbf{w})$, which represents the average gradient direction in the segment.

## 3   SAMPLE-WISE GRADIENTS AT INITIALIZATION FOR BINARY CLASSIFICATION

In this section, we analyze the directions of sample-wise gradients at randomly initialization of neural networks for binary classification, and show that the noisy segment gradients are opposite in direction to the clean segment gradients.

Considering the close relation between the sample-wise gradient $\nabla l_i(\mathbf{w})$ and $\nabla h(\mathbf{w}; \mathbf{x}_i)$ (see Eq.(4)), we start with the model derivative $\nabla h$.

**Direction of the model derivative $\nabla h$.** Given two arbitrary inputs $\mathbf{x}, \mathbf{z} \in \mathbb{R}^d$, we denote the angle between $\mathbf{x}$ and $\mathbf{z}$ in the input space $\mathbb{R}^d$ by $\theta_d(\mathbf{x}, \mathbf{z})$, and denote the angle between the two model derivative vectors $\nabla h(\mathbf{w}_0; \mathbf{x})$ and $\nabla h(\mathbf{w}_0; \mathbf{z})$ by $\theta_h(\mathbf{x}, \mathbf{z})$. The first observation is that: *similar inputs (relatively small angle $\theta_d$) have similar model derivatives (relatively small angle $\theta_h$)*, as formalized in the following theorem (see proof in Appendix A.1).

**Theorem 3.1.** *Consider an infinitely wide neural network $h$ as defined in Eq.(2) at random initialization $\mathbf{w}_0$. The followings hold:*

1. *given two inputs $\mathbf{x}$ and $\mathbf{z}$, if $\theta_d(\mathbf{x}, \mathbf{z}) \ll 1$, then $\theta_h(\mathbf{x}, \mathbf{z}) \ll 1$;*

2. *for any three inputs $\mathbf{x}$, $\mathbf{z}$ and $\mathbf{z}'$, if $0 \leq \theta_d(\mathbf{x}, \mathbf{z}) \leq \theta_d(\mathbf{x}, \mathbf{z}') \leq \frac{\pi}{2}$, then $0 \leq \theta_h(\mathbf{x}, \mathbf{z}) \leq \theta_h(\mathbf{x}, \mathbf{z}') \leq \frac{\pi}{2}$.*

We experimentally observe that the same relation also hold on neural network with finite width. See Figure 2.

Hence, the map $\nabla h : \mathbf{x} \mapsto \nabla h(\mathbf{x})$, from input space to the space of model derivatives, is expected to preserve the cluster structures. As illustrated in Figure 3, the two clusters of data inputs are mapped to two clusters of model derivative vectors: data pairs from the same cluster ("within") still have similar model derivatives (small $\theta_h$), while cross-cluster pairs ("between") have large $\theta_h$. The implementation details for Figure 2 and Figure 3 can be found in Appendix B.1.

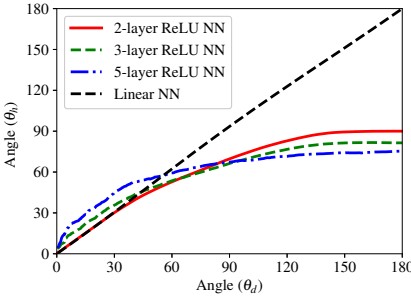

Figure 2: Relation between $\theta_h$ and $\theta_d$. Similar inputs (small $\theta_d$) implies similar model derivatives (small $\theta_h$).

**Directions of the sample-wise gradients.** By Eq.(4), the direction of sample-wise gradient $\nabla l(\mathbf{w}_0)$ is mostly determined by $\nabla h(\mathbf{w}_0; \mathbf{x})$, with the scalar $y - f(\mathbf{w}_0; \mathbf{x})$ only controls the sign.

Consider the subset of data from the same ground truth class, $\mathcal{D}^{(c)} = \mathcal{D}_{clean}^{(c)} \cup \mathcal{D}_{noise}^{(c)}$. Noting that the segments $\mathcal{D}_{noise}^{(c)}$ and $\mathcal{D}_{clean}^{(c)}$ have the same data distribution, they also have the same $\nabla h$ distribution. However, the two segments have different labels, either 0 or 1. Recalling that $0 < f(\mathbf{w}_0; \mathbf{x}) < 1$, we see that the scalar $y - f(\mathbf{w}_0; \mathbf{x})$ have opposite signs for the two segments. Hence, for each sample in $\mathcal{D}_{noise}^{(c)}$, the label corruption flips the sample-wise gradient to the opposite direction, as $y - f(\mathbf{w}_0; \mathbf{x})$ and $\hat{y} - f(\mathbf{w}_0; \mathbf{x})$ have different signs.

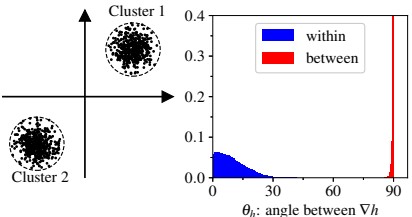
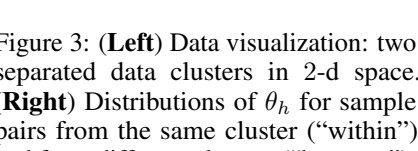
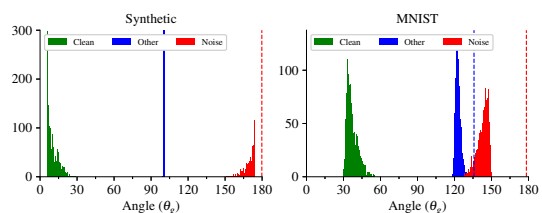

Figure 3: (**Left**) Data visualization: two separated data clusters in 2-d space. (**Right**) Distributions of $\theta_h$ for sample pairs from the same cluster ("within") and from different clusters ("between").

Figure 4: The distributions of $\theta_g$. **Left**: synthetic data in Figure 3 ($\delta = 0.3$), **Right**: two classes MNIST (("0" and 1", $\delta = 0.3$). Dash lines represent segment gradients. Both cases use a 2-layer ReLU neural network.

This flip results in a split in the sample-wise gradient distributions between $\mathcal{D}_{noise}^{(c)}$ and $\mathcal{D}_{clean}^{(c)}$. Presumably, inputs from the same ground truth class tend to be more similar (with small angles $\theta_d$), compared to others. By the analysis for $\nabla h$, we expect the angles $\theta_h$ within $\mathcal{D}^{(c)}$ are also relatively small. Let's denote the angles between a pair of sample-wise gradients by $\theta_g$. Within each segment, either $\mathcal{D}_{noise}^{(c)}$ or $\mathcal{D}_{clean}^{(c)}$, $\theta_g$ keeps relatively small. However, for cross segment pairs, $\theta_g$ becomes close to $180°$ due to the flip. As a consequence, the segment gradients $g_{noise}^{(c)}(\mathbf{w}_0)$ and $g_{clean}^{(c)}(\mathbf{w}_0)$ are sharply opposite to each other.

Figure 4 experimentally confirms this expectation [2] . We see that the distributions of $\mathcal{G}_{noise}^{(c)}(\mathbf{w}_0)$ and $\mathcal{G}_{clean}^{(c)}(\mathbf{w}_0)$ are well-separated and symmetrically located. More importantly, the angle $\theta_g$ between clean and noisy segment gradients (red dash line) is close to $180°$, meaning a sharp opposition between the two segment gradients. In this figure, we add an "other" segment, which represents data that is not in $\mathcal{D}^{(c)}$, for reference.

**Magnitudes of segment gradients.** We are interested in the magnitudes of $g_{clean}^{(c)}(\mathbf{w}_0)$ and $g_{noise}^{(c)}(\mathbf{w}_0)$. By definition, for $k \in \{clean, noise\}$,
$$g_k^{(c)}(\mathbf{w}_0) = |D_k^{(c)}|\mathbb{E}[\nabla l] = |D_k^{(c)}|\mathbb{E}[f(\mathbf{w}_0; \mathbf{x}) - y]\mathbb{E}[\nabla h]$$
where the expectation is taken over the corresponding data segment. We know that $\mathbb{E}[\nabla h]$ is the same for clean and noisy segments. In addition, $\mathbb{E}[f(\mathbf{w}_0; \mathbf{x}) - y]$ are opposite for these two segments, as $\mathbb{E}[f(\mathbf{w}_0; \mathbf{x})]$ is 0.5 by random guess and $y = 1$ for one segment and $y = 0$ for the other. Hence, we see that the magnitudes $\|g_k^{(c)}\|$ are determined by the segment population, and we have
$$\|g_{clean}^{(c)}(\mathbf{w}_0)\|/\|g_{noise}^{(c)}(\mathbf{w}_0)\| = (1 - \delta)/\delta > 1. \tag{5}$$

## 4 LEARNING DYNAMICS OF BINARY CLASSIFICATION

In this section, we analyze the learning dynamics of binary classification with label noise in the training dataset.

Specifically, we show that in the early stage of training, the dynamics exhibits a clean-priority learning characteristic, due to a dominance of the clean segment in first-order information, i.e., sample-wise gradients. We further show that in later stage of training, this dominance fades away and clean-priority learning terminates, resulting in a fitting of the noisy samples and worsening of the test performance.

We partition the optimization procedure into two stages: early stage which happens before the early stopping point; and later stage which is after the early stopping point.

### 4.1 INITIALIZATION & EARLY STAGE

In Section 3, we have seen that, at initialization,
$$g_{noise}^{(c)}(\mathbf{w}_0) = -\alpha_0 g_{clean}^{(c)}(\mathbf{w}_0), \tag{6}$$

---

[2]For illustration purpose, we compare each sample-wise gradient with $g_{clean}^{(c)}(\mathbf{w}_0)$.

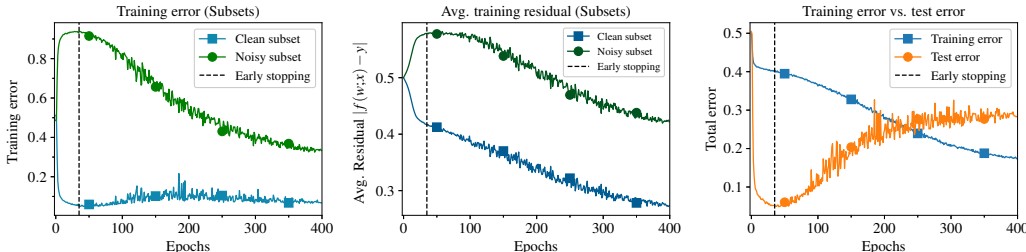

Figure 5: Learning dynamics on two classes ("7" and "9") of MNIST ($\delta = 0.4$) with FCN. **Left**: clean segment error vs. noisy segment error. **Middle**: average residuals on clean and noisy segments. **Right**: total test error and total training error. Vertical dash line represents the early stopping point.

with $\alpha_0 \triangleq \delta/(1-\delta) \in (0,1)$. We note that, during training, the model derivative $\nabla h$ for an infinitely wide neural network is found to be unchanged (Liu et al., 2020):

$$\nabla h(\mathbf{w}_t) = \nabla h(\mathbf{w}_0), \ \forall t > 0, \ \text{when } m \to \infty.$$

By Eq.(4), this implies that each sample-wise gradient $\nabla l_i$ keeps its direction unchanged during training (but changes in magnitude through the factor $f(\mathbf{w}; \mathbf{x}) - y$). Therefore, it is reasonable to make the following assumption:

**Assumption 4.1.** There exist a time $T > 0$ and a sequence $\{\alpha_t\}_{t=0}^T$, with each $\alpha_t \in (0,1)$, such that, for all $t \in [0, T]$ and $c \in \{0, 1\}$, the following holds $g_{noise}^{(c)}(\mathbf{w}_t) = -\alpha_t g_{clean}^{(c)}(\mathbf{w}_t)$.

Define $\hat{g}^{(c)}(\mathbf{w})$ as the summation of the sample-wise gradients with ground truth labels, i.e., $\hat{g}^{(c)}(\mathbf{w}) = \sum_{(\mathbf{x}, \hat{y}) \in \hat{\mathcal{D}}^{(c)}} \nabla l(\mathbf{w}; \mathbf{x}, \hat{y})$. By the assumption, we have for all $0 \le t \le T$ and $c \in \{0, 1\}$,

$$g_{clean}^{(c)}(\mathbf{w}_t) = \frac{1}{\alpha_t + 1} \hat{g}^{(c)}(\mathbf{w}_t). \tag{7}$$

On the other hand, by definition, we have for full gradient

$$\nabla L(\mathbf{w}_t; \mathcal{D}) = \sum_c \left( g_{clean}^{(c)}(\mathbf{w}_t) + g_{noise}^{(c)}(\mathbf{w}_t) \right). \tag{8}$$

Combining Assumption 4.1 and Eqs.(7) and (8), we easily have the following lemma:

**Lemma 4.2** (Update rules). *Suppose Assumption 4.1 holds with time $T > 0$ and sequence $\{\alpha_t\}_{t=0}^T \in (0,1)^T$. Then, the gradient descent (with learning rate $\eta$) have the following update rule*

$$\mathbf{w}_{t+1} = \mathbf{w}_t - \eta_t' \nabla L(\mathbf{w}_t; \hat{\mathcal{D}}), \ \text{for } t \le T, \tag{9}$$

*with $\eta_t' = \frac{1-\alpha_t}{1+\alpha_t} \eta > 0$.*

**Remark 4.3** (mini-batch scenario). In mini-batch SGD, similar relation of Eq.(9) also holds for a mini-batch estimation $\nabla L(\mathbf{w}_0; \mathcal{B})$, as long as the sampling of the mini-batch is independent of the label noise and the batch size $|\mathcal{B}|$ is not too small such that the majority of samples are clean in the batches. Hence, in the following, we do not explicitly write out the dependence on the mini-batches.

The theorem states that, after adding label noise to the training dataset, the gradient descent update is equivalent to the one without label noise (except a different learning rate $\eta_t' < \eta$). In another word, the gradient descent does not essentially "see" the noisy data and its update direction is determined only by the clean samples.

**Clean-priority learning.** This theorem implies the following learning characteristics of what we call *clean-priority learning*, as we described below.

*Segment training loss and accuracy.* The loss $L(\mathbf{w}; \mathcal{D}_{clean})$ on the clean segment keeps decreasing, while the loss $L(\mathbf{w}; \mathcal{D}_{noise})$ on the noisy segment is increasing, as formally stated in the following Theorem (see the proof in Appendix A.3):

**Theorem 4.4.** *Suppose Assumption 4.1 holds with time $T > 0$ and sequence $\{\alpha_t\}_{t=0}^T$, $\alpha_t \in (0,1)$. We have, for all $t \in [0,T]$ and sufficiently small $\eta$,*

$$L(\mathbf{w}_{t+1}; \mathcal{D}_{clean}) < L(\mathbf{w}_t; \mathcal{D}_{clean});$$
$$L(\mathbf{w}_{t+1}; \mathcal{D}_{noise}) > L(\mathbf{w}_t; \mathcal{D}_{noise}).$$

Accordingly, the training accuracy on the clean segment is increased, and that on the noisy segment is decreased.

*Residual magnitude:* $|f(\mathbf{w}; \mathbf{x}) - y|$. As a consequence of the decreasing clean segment loss $L(\mathbf{w}; \mathcal{D}_{clean})$, the clean training samples are learned, in the sense that the network output $f(\mathbf{w}; \mathbf{x})$ moves towards its corresponding label $y$, i.e., $|f(\mathbf{w}; \mathbf{x}) - y|$ decreases on the clean segment. On the other hand, the increase of the $L(\mathbf{w}; \mathcal{D}_{noise})$ results in that, on the noisy segment, the network output $f(\mathbf{w}; \mathbf{x})$ moves away from its corresponding label $y$, but towards its ground truth label $\hat{y}$. Namely, the noisy segment is not learnt.

*Test loss.* As the test dataset $\mathcal{D}_{test}$ is not label-corrupted and is drawn from the same data distribution as $\hat{\mathcal{D}}$, it is expected that the update rule in Eq.(9) decreases the test loss $L(\mathbf{w}; \mathcal{D}_{test})$.

Figure 5 shows the clean-priority learning phenomenon on a binary classification of two classes of MNIST. The relevant part is the early stage, i.e., before the early stopping point (left of the vertical dash line). As one can see, in this stage, the prediction error and noisy segment loss $L(\mathbf{w}; \mathcal{D}_{noise})$ keep increasing (See Appendix B for segment loss curves). Especially, the prediction error increases from a random guess (error $= 0.5$) at initialization towards $100\%$. Meanwhile, the clean segment loss and prediction error keep decreasing. Moreover, the average residual magnitude $|f(\mathbf{w}; \mathbf{x}) - y|$ decreases on the clean segment, but increases on the noisy segment, implying that only clean segment is learnt. These behaviors illustrate that in the early stage the learning dynamics prioritize the clean samples.

In short, in the early stage, the *clean-priority learning* prioritizes the learning on clean training samples. The interesting point is that, although it seems impossible to distinguish the clean from the noisy directly from the data, this prioritization is possible because the model have access to the first-order information, i.e., sample-wise gradients. Importantly, it is this awareness of the clean samples and this prioritization in the early stage that allow the possibility of achieving test performances better than the noisy level.

## 4.2 EARLY STOPPING POINT & LATER STAGE

As we have seen in the above subsection, the dominance of the magnitude $\|g_{clean}^{(c)}\|$ over $\|g_{noise}^{(c)}\|$ is one of the key reasons to maintain the clean-priority learning in the early stage. However, we shall see shortly that this dominance diminish as the training goes on, resulting in a final termination of the clean-priority learning.

**Diminishing dominance of the clean gradient.** Recall that the sample-wise gradient is proportional to the magnitude of the residual: $\nabla l(\mathbf{w}) \propto y - f(\mathbf{w}; \mathbf{x})$. The learning of a sample, i.e., decreased $|y - f(\mathbf{w}; \mathbf{x})|$, results in a decrease in the magnitude $|\nabla l(\mathbf{w})|$. As an effect of the clean-priority learning, the residuals magnitude $|f(\mathbf{w}; \mathbf{x}) - y|$ evolves differently for different data segments: *decreases* on the clean segment $\mathcal{D}_{clean}$, but *increases* on the noisy segment $\mathcal{D}_{noise}$. This difference leads to the diminishment of the dominance of clean segment $\|g_{clean}^{(c)}(\mathbf{w})\|$, which originates from the dominance of the population of clean training samples.

**Theorem 4.5** (Diminishing dominance of the clean gradient). *Assume the neural network is infinitely wide and the learning rate $\eta$ of the gradient descent is sufficiently small. Suppose Assumption 4.1 holds with time $T > 0$ and sequence $\{\alpha_t\}_{t=0}^T \in (0,1)^T$. The sequence $\{\alpha_t\}_{t=0}^T$ monotonically increases: for all $t \in [0,T]$, $\alpha_{t+1} > \alpha_t$.*

Please find the proof in Appendix A.4. As $\alpha_t$ measures this clean dominance ($\alpha_t$ close to 1 means less dominant), this theorem indicates that the dominance diminishes as the training goes on.

Figure 6 illustrates this diminishing dominance on the two class MNIST classification problem. In the early stage, the ratio $\|g_{clean}^{(c)}(\mathbf{w})\|/\|g_{noise}^{(c)}(\mathbf{w})\|$ starts with a value around the ratio of population

$(1 - \delta)/\delta = 1.5$, and monotonically decrease to around 1 at or before the early stopping point, indicating that the dominance vanishes.

**Learning the noisy samples.** In the later stage (i.e., after the early stopping point), the magnitudes of $\|g_{clean}^{(c)}(\mathbf{w})\|$ and $\|g_{noise}^{(c)}(\mathbf{w})\|$ are similar, and there is no apparent dominance of one over the other. Then, the model and algorithm do not distinguish the clean segment from the noisy one, and there will be no clean-priority learning. In this stage, the model learns both the clean and noisy segments, aiming at achieving exact fitting of the training data. Ultimately, training errors of both segments converge to zero.

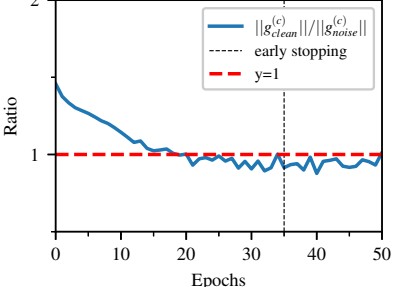

Figure 6: Diminishing dominance of clean gradient.

It is expected that in this stage the loss and prediction error on the test dataset $\mathcal{D}_{test}$ become worse, as the learning on the noisy segment contaminates the performance achieved by the clean-priority learning in the earlier stage.

As illustrated in Figure 5, after the early stopping point, the noisy segment starts to be learnt. Specifically, both training loss and error on this segment turn to decrease towards zero; the average residual magnitude $|f(\mathbf{w}; \mathbf{x}) - y|$ turn to decrease, indicating that the network output $f(\mathbf{w}; \mathbf{x})$ is learnt to move towards its (corrupted) label. It is worth to note that the learning on the clean segment is still ongoing, as both training loss and error on this segment keeps decreasing.

In high level, before the first stage, the learning procedure prioritizes the clean training samples, allowing the superior-noise-level performance on the test dataset; in later stage, the learning procedure picks up the noisy samples, worsening the test performance toward the noise-level.

## 5 MULTI-CLASS CLASSIFICATION

In this section we show that multi-class classification problems exhibit the same learning dynamics as described in Section 4. For multi-class classification, we consider a variant of the sample-wise gradient, *single-logit sample-wise gradient*.

**Single-logit sample-wise gradients.** In a $C$-class classification problem, the neural network $f$ has $C$ output logits, and the labels are a $C$-dimensional one-hot encoded vectors. One can view the neural network as $C$ co-existing binary classifiers. Specifically, for each $c \in \{1, 2, \cdots, C\}$, the $c$-th logit $f_c$ is a binary classifier, and the $c$-th component of the label $y_c \in \{0, 1\}$ is the binary label for $f_c$.

By Eq.(4), the sample-wise gradient can be written as $\nabla l(\mathbf{w}) = \sum_{c=1}^{C} \nabla l_c(\mathbf{w})$, where

$$\nabla l_c(\mathbf{w}) \triangleq (f_c(\mathbf{w}; \mathbf{x}) - y_c) \nabla h_c(\mathbf{w}; \mathbf{x}) \tag{10}$$

is the *single-logit sample-wise gradient*, which only depends on the corresponding single logit.

We point out that, the cleanness of a sample is only well defined with respect to each single logit, but not to the whole output. For example, consider a sample with ground truth label 0 but is incorrectly labeled as class 1. For all the rest binary classifiers, except the 0-th and 1-st, this sample is always considered as the negative class, as $y_c = 0$ for all $c \neq 0, 1$; hence, the noisy sample is considered "clean", for these $C - 2$ binary classifiers. Therefore, a noisy sample is not necessarily noisy for all the $C$ binary classifiers.

With this observation, we consider the single-logit sample-wise gradient $\nabla l_c(\mathbf{w})$ instead.

**At initialization.** Given $c \in \{1, 2, \cdots, C\}$, the $c$-logit sub-network $h_c$ (before *softmax*) is the same as the network $h$ discussed in Section 3, and the output $f_c \in (0, 1)$. Hence, all the directional analysis for binary case (Section 3) still applies to the single-logit sample-wise gradient $\nabla l_c(\mathbf{w})$. See Appendix C for numerical verification.

Different from the *sigmoid* output activation which tends to predict an average of 0.5 before training, the *softmax* has an average output $f_c$ around $1/C$ with random guess at initialization. This leads to

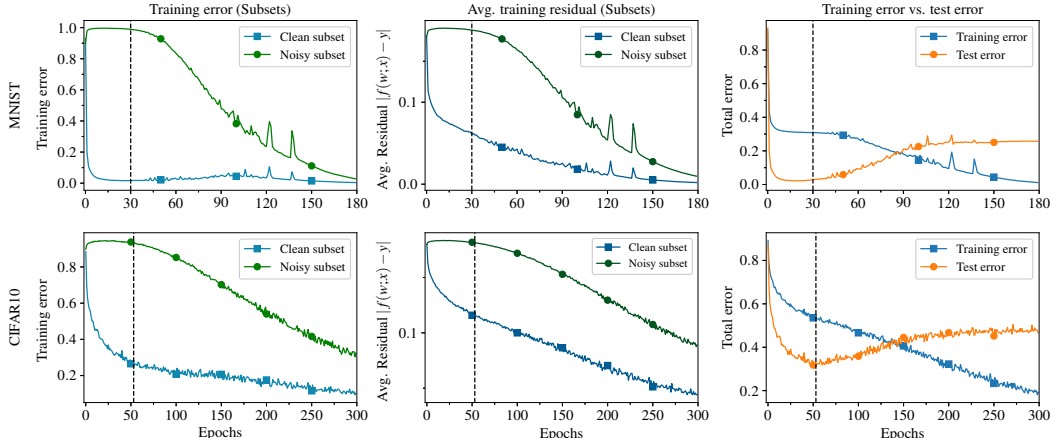

Figure 7: Learning dynamics on multi-class classification. **Left**: clean segment error vs. noisy segment error. **Middle**: average residuals on clean and noisy segments. **Right**: total test error and total training error. Vertical dash line represents the early stopping point.

$\mathbb{E}|f_c(\mathbf{w}_0; \mathbf{x}) - y_c| = 1 - 1/C$ when $y_c = 1$, and $\mathbb{E}|f_c(\mathbf{w}_0; \mathbf{x}) - y_c| = 1/C$ when $y_c = 0$. Recalling that $\mathcal{D}_{clean}^{(c)}$ and $\mathcal{D}_{noise}^{(c)}$ (hence the corresponding $\nabla h$) have the same distribution, using Eq.(10) we have

$$g_{noise}^{(c)}(\mathbf{w}_0) \approx -\delta \hat{g}^{(c)}(\mathbf{w}_0)/(C-1), \tag{11a}$$

$$g_{clean}^{(c)}(\mathbf{w}_0) \approx (1-\delta)\hat{g}^{(c)}(\mathbf{w}_0). \tag{11b}$$

Therefore, we have the dominance of $\|g_{clean}^{(c)}\|$ over $\|g_{noise}^{(c)}\|$ at initialization, with a ratio

$$\|g_{clean}^{(c)}(\mathbf{w}_0)\|/\|g_{clean}^{(c)}(\mathbf{w}_0)\| \approx (C-1)(1-\delta)/\delta.$$

**Learning dynamics.** As the configuration of $\nabla l_c$ is similar to that of a binary classification, we expect similar learning dynamics as discussed in Section 4, especially the clean-priority learning, happen for multi-class classification.

We conduct experiments to classify the MNIST (with added label noise $\delta = 0.3$) and CIFAR-10 (with added label noise $\delta = 0.4$) datasets using a CNN and a ResNet, respectively. As

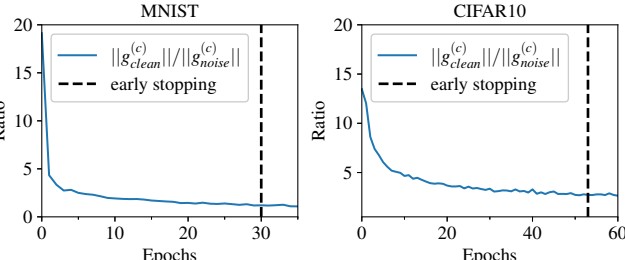

Figure 8: Diminishing dominance of the clean gradient on multi-class classification.

is shown in Figure 7 and Figure 8, in most of the early stage, the clean segment has clean dominance over the noise segment and the dynamics shows the clean-priority learning characteristic, decreasing the clean segment error and residual, but increasing the noisy segment error and residual. Furthermore, the dominance of the clean segment monotonically decreases until the early stopping point. In the later stage, the networks start to learn the noisy segments. See the experimental setup in Appendix B.

## 6 CONCLUSION

In this paper, we delved into the mechanism and dynamics of clean-priority learning by analyzing sample-wise gradients of infinitely wide networks. We demonstrate that label noise flips a sample-wise gradient to its opposite direction, causing the noisy sample-wise gradients being cancelled out by the clean ones in the early stages of training. Consequently, the gradient descent update direction aligns with that of the clean segment gradient, allowing the early learning of clean samples. Moreover, as clean-priority learning progresses, the dominance of the clean sample-wise gradients gradually diminishes, leading to a termination of the clean-priority learning around the early stopping point.

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
