# A  TECHNICAL PROOFS

## A.1  PROOF OF THEOREM 3.1

We restate Theorem 3.1 below for easier reference.

**Theorem A.1** (Theorem 3.1). *Consider an infinitely wide neural network $h$ as defined in Eq.(2) at random initialization $\mathbf{w}_0$. The followings hold:*

1. *given two inputs $\mathbf{x}$ and $\mathbf{z}$, if $\theta_d(\mathbf{x}, \mathbf{z}) \ll 1$, then $\theta_h(\mathbf{x}, \mathbf{z}) \ll 1$;*

2. *for any three inputs $\mathbf{x}$, $\mathbf{z}$ and $\mathbf{z}'$, if $0 < \theta_d(\mathbf{x}, \mathbf{z}) < \theta_d(\mathbf{x}, \mathbf{z}') < \frac{\pi}{2}$, then $0 < \theta_h(\mathbf{x}, \mathbf{z}) < \theta_h(\mathbf{x}, \mathbf{z}') < \frac{\pi}{2}$.*

*Proof.* First of all, we introduce some useful notations. We rewrite the (pre-activated) neural network $h$ as

$$
\begin{aligned}
\alpha^{(0)}(\mathbf{x}) &= \mathbf{x} \\
\alpha^{(l)}(\mathbf{x}) &= \frac{\sqrt{2}}{\sqrt{m_l}} \sigma \left( W^{(l)} \alpha^{(l-1)}(\mathbf{x}) \right), \quad \forall l \in \{1, 2, \cdots, L\}, \\
h(\mathbf{x}) &= W^{(L+1)} \alpha^{(L)}(\mathbf{x}).
\end{aligned}
\tag{12}
$$

Given any two inputs $\mathbf{x}, \mathbf{z} \in \mathbb{R}^d$ and $l \in \{0, 1, \cdots, L\}$, we also denote $\theta^{(l)}(\mathbf{x}, \mathbf{z})$ as the angle between the $l$-th layer vectors $\alpha^{(l)}(\mathbf{x})$ and $\alpha^{(l)}(\mathbf{z})$, i.e., $\theta^{(l)}(\mathbf{x}, \mathbf{z}) \triangleq \arccos \left( \frac{\langle \alpha^{(l)}(\mathbf{x}), \alpha^{(l)}(\mathbf{z}) \rangle}{\|\alpha^{(l)}(\mathbf{x})\| \|\alpha^{(l)}(\mathbf{z})\|} \right)$.

It was shown by Liu & Hui (2023) that the angle $\theta_h$ has the following relation with $\theta^{(l)}$ and $\theta_d$.

**Theorem A.2** (modified from Liu & Hui (2023)). *Consider the infinitely wide ReLU neural network $h$ of depth $L$, as defined in Eq.(2), at random initialization $\mathbf{w}_0$. Given two arbitrary inputs $\mathbf{x}$ and $\mathbf{z}$, the angle $\theta_h(\mathbf{x}, \mathbf{z})$ satisfies*

$$
\cos \theta_h(\mathbf{x}, \mathbf{z}) = \frac{1}{L+1} \sum_{l=0}^{L} \left[ \cos \theta^{(l)}(\mathbf{x}, \mathbf{z}) \prod_{l'=l}^{L-1} (1 - \theta^{(l')}(\mathbf{x}, \mathbf{z})/\pi) \right].
\tag{13}
$$

*Moreover, if define function $g : [0, \pi) \to [0, \pi)$ as $g(z) = \arccos \left( \frac{\pi - z}{\pi} \cos z + \frac{1}{\pi} \sin z \right)$ and let $g^l(\cdot)$ be the $l$-fold composition of $g(\cdot)$, then*

$$
\theta_h^{(l)}(\mathbf{x}, \mathbf{z}) = g^l \left( \theta_d(\mathbf{x}, \mathbf{z}) \right).
\tag{14}
$$

*Furthermore, if $\theta_d(\mathbf{x}, \mathbf{z}) \ll 1$, then*

$$
\cos \theta_h(\mathbf{x}, \mathbf{z}) = \left( 1 - \frac{L}{2\pi} \theta_d + o(\theta_d) \right) \cos \theta_d.
\tag{15}
$$

When $\theta_d(\mathbf{x}, \mathbf{z}) \ll 1$, by Eq.(15), we have

$$
1 - \cos \theta_h(\mathbf{x}, \mathbf{z}) = \frac{L}{2\pi} \theta_d(\mathbf{x}, \mathbf{z}) + o(\theta_d) \ll 1.
\tag{16}
$$

This directly implies that $\theta_h(\mathbf{x}, \mathbf{z}) \ll 1$. We conclude the first statement of the theorem.

The second statement relies on the following lemma about the properties of function $g$.

**Lemma A.3.** *The function $g : [0, \pi) \to [0, \pi)$ as defined in Theorem A.2 satisfies the following:*

- *if $z \in (0, \frac{\pi}{2})$, then $g(z) \in (0, \frac{\pi}{2})$.*

- *function $g$ is monotonically increasing in $(0, \frac{\pi}{2})$.*

Applying the above lemma onto Eq.(14), we have, when $0 < \theta_d(\mathbf{x}, \mathbf{z}) < \theta_d(\mathbf{x}, \mathbf{z}') < \frac{\pi}{2}$,

$$
0 < \theta^{(l)}(\mathbf{x}, \mathbf{z}) < \theta^{(l)}(\mathbf{x}, \mathbf{z}') < \frac{\pi}{2}, \quad \forall l \in \{1, 2, \cdots, L\}.
\tag{17}
$$

Applying this relation to Eq.(13), we get

$$
\cos \theta^{(l)}(\mathbf{x}, \mathbf{z}) > \cos \theta^{(l)}(\mathbf{x}, \mathbf{z}') > 0.
\tag{18}
$$

Therefore, we have $0 < \theta_h(\mathbf{x}, \mathbf{z}) < \theta_h(\mathbf{x}, \mathbf{z}') < \frac{\pi}{2}$. $\qquad\square$

## A.2 Proof of Lemma A.3

*Proof.* By definition of $g$,

$$\cos g(z) = \frac{\pi - z}{\pi} \cos z + \frac{1}{\pi} \sin z. \tag{19}$$

It is easy to see that, when $z \in (0, \frac{\pi}{2})$, the right hand side of the above equation is positive. Therefore, $g(z)$ stays in $)0, \frac{\pi}{2})$.

As for the monotonicity, we first investigate the R.H.S. of Eq.(19). We have

$$\frac{d(\text{R.H.S.})}{dz} = -\left(1 - \frac{z}{\pi}\right) \sin z, \tag{20}$$

which is always negative for $z \in (0, \pi/2)$. Combining the monotonicity of $\arccos(\cdot)$, we have the conclusion. □

## A.3 Proof of Theorem 4.4

*Proof.* First, note that the clean data has the same distribution as the noiseless (ground-truth-labelled) data. Hence, $L(\mathbf{w}; \mathcal{D}_{clean}) = L(\mathbf{w}; \hat{\mathcal{D}})$. By Lemma 4.2, the gradient descent minimizes $L(\mathbf{w}; \hat{\mathcal{D}})$, as long as the learning rate $\eta$ is small enough to avoid over-shooting. Therefore, it is straightforward to get that the gradient descent also decreases the clean segment loss $L(\mathbf{w}; \mathcal{D}_{clean})$.

Let's consider the noisy segment $\mathcal{D}_{noise}$. Combining Assumption 4.1 and Eqs.(7) and (8), we get

$$\nabla L(\mathbf{w}_t; \hat{\mathcal{D}}) = -\frac{1 - \alpha_t}{\alpha_t} \sum_c g_{noise}^{(c)}(\mathbf{w}_t) = -\frac{1 - \alpha_t}{\alpha_t} \nabla L(\mathbf{w}_t, \mathcal{D}_{noise}). \tag{21}$$

We note that the factor $-\frac{1-\alpha_t}{\alpha_t}$ is negative, indicating that the gradient descent update, Eq.(9), is in opposite direction of minimizing the noisy segment $L(\mathbf{w}_t, \mathcal{D}_{noise})$. Hence, we get that $L(\mathbf{w}_{t+1}, \mathcal{D}_{noise}) > L(\mathbf{w}_t, \mathcal{D}_{noise})$. □

## A.4 Proof of Theorem 4.5

*Proof.* First note that an infinitely wide feedforward neural network (before the activation function on output layer) is linear in its parameters, and can be written as (Liu et al., 2020; Zhu et al., 2022):

$$h(\mathbf{w}; \mathbf{x}) = h(\mathbf{w}_0; \mathbf{x}) + \nabla h(\mathbf{w}_0; \mathbf{x})^T (\mathbf{w} - \mathbf{w}_0), \tag{22}$$

where $\nabla h(\mathbf{w}_0; \mathbf{x})$ is constant during training. As is known, the logistic regression loss (for an arbitrary $\mathcal{S}$) on a linear model

$$L(\mathbf{w}; \mathcal{S}) = \sum_{(\mathbf{x}, y) \in \mathcal{S}} -y \log f(\mathbf{w}; \mathbf{x}) - (1 - y) \log(1 - f(\mathbf{w}; \mathbf{x})), \tag{23}$$

is a convex function with respect to the parameters $\mathbf{w}$, where $f(\mathbf{w}; \mathbf{x}) = sigmoid(h(\mathbf{w}; \mathbf{x})) = 1/(1 + \exp(-h(\mathbf{w}; \mathbf{x})))$. Hence, at any point $\mathbf{w}$ we have the Hessian matrix $H(\mathbf{w}; \mathcal{S})$ of the logistic regression loss $L(\mathbf{w}; \mathcal{S})$ is positive definite.

Now, consider the point $\mathbf{w}_{t+1} = \mathbf{w}_t - \eta L(\mathbf{w}_t; \mathcal{D})$ with a sufficiently small step size $\eta$. Using Assumption 4.1 and Eq.(8), we can also write $\mathbf{w}_{t+1}$ as

$$\mathbf{w}_{t+1} = \mathbf{w}_t - \eta(1 - \alpha_t) L(\mathbf{w}_t; \mathcal{D}_{clean}), \text{ or}$$

$$\mathbf{w}_{t+1} = \mathbf{w}_t + \eta \frac{1 - \alpha_t}{\alpha_t} L(\mathbf{w}_t; \mathcal{D}_{noise}).$$

For $\mathcal{D}_{clean}$, we have

$$\nabla L(\mathbf{w}_{t+1}; \mathcal{D}_{clean}) = \nabla L(\mathbf{w}_t; \mathcal{D}_{clean}) + H(\xi; \mathcal{D}_{clean})(\mathbf{w}_{t+1} - \mathbf{w}_t), \tag{24}$$

with $\xi$ being some point between $\mathbf{w}_t$ and $\mathbf{w}_{t+1}$. Then,

$$\|\nabla L(\mathbf{w}_{t+1}; \mathcal{D}_{clean})\|^2 = \|\nabla L(\mathbf{w}_t; \mathcal{D}_{clean})\|^2 - 2\eta(1 - \alpha_t) \nabla L(\mathbf{w}_t; \mathcal{D}_{clean})^T H(\xi; \mathcal{D}_{clean}) \nabla L(\mathbf{w}_t; \mathcal{D}_{clean}) + O(\eta^2).$$

By the convexity of the loss function (i.e., the positive definiteness of Hessian $H$), we easily get

$$\|\nabla L(\mathbf{w}_{t+1}; \mathcal{D}_{clean})\|^2 < \|\nabla L(\mathbf{w}_t; \mathcal{D}_{clean})\|^2.$$

Similarly for $\mathcal{D}_{noise}$,

$$\|\nabla L(\mathbf{w}_{t+1}; \mathcal{D}_{noise})\|^2 = \|\nabla L(\mathbf{w}_t; \mathcal{D}_{noise})\|^2 + 2\eta \frac{1-\alpha_t}{\alpha_t} \nabla L(\mathbf{w}_t; \mathcal{D}_{noise})^T H(\xi'; \mathcal{D}_{noise}) \nabla L(\mathbf{w}_t; \mathcal{D}_{noise}) + O(\eta^2).$$

Hence, for small $\eta$, we get

$$L(\mathbf{w}_{t+1}; \mathcal{D}_{noise})\|^2 > \|\nabla L(\mathbf{w}_t; \mathcal{D}_{noise})\|^2.$$

Noting that

$$\nabla L(\mathbf{w}_t; \mathcal{D}_{noise}) = \sum_c g_{noise}^{(c)}(\mathbf{w}_t) = -\alpha_t \sum_c g_{clean}^{(c)}(\mathbf{w}_t) = -\alpha_t \nabla L(\mathbf{w}_t; \mathcal{D}_{clean}),$$

we obtain

$$\alpha_{t+1} > \alpha_t. \tag{25}$$

Therefore, we conclude the proof of the theorem. □

The high-level idea of the above proof is that: (locally) decreasing a convex function $L$ along the opposite gradient direction, $-\nabla L$, results in shrinking the magnitude of the gradient; (locally) increasing a convex function $L$ along the gradient direction $\nabla L$ results in magnifying the gradient magnitude.

## B  EXPERIMENTAL SETUP DETAILS

### B.1  EXPERIMENTAL SETUP FOR FIGURE 2 AND 3

For the experiment in Figure 2, we consider six neural networks: three linear networks with 2, 3 and 5 layers, respectively; and three ReLU networks with 2, 3 and 5 layers. Each hidden layer of each neural network has 512 neurons. For each network, we compute the model derivatives $\nabla h$ on the 1-sphere $\mathcal{S}^1 = \{(\cos\theta_d, \sin\theta_d) : \theta_d \in [0, 2\pi)\}$, at the network initialization. Figure 2 shows the relations between the angle $\theta_h$ and $\theta_d$.[3]

For Figure 3, we consider the following synthetic dataset (also shown in the left panel of Figure 3): two separated data clusters in a 2-dimensional space. We use a 3-layer ReLU network of width $m = 512$ at its initialization to compute the sample-wise model derivatives $\nabla h$. The right panel of Figure 3 shows the distributions of angle $\theta_h$ for data pairs from the same cluster ("within") and from different clusters ("between"). It can be easily seen that the "within" distribution has smaller angles $\theta_h$ than the "between" distribution, which is expected as the data from the same clusters are more similar.

### B.2  EXPERIMENTAL SETUP FOR CLASSIFICATION PROBLEMS

**Binary classification on two class of MNIST.**  We extract two classes, the images with digits "7" and "9", out from the MNIST datasets, and injected 30% random label noise into each class in the training dataset (i.e., labels of 30% randomly selected samples are flipped to the other class), leaving test set intact. We employ a fully connected neural network with 2 hidden layers, each containing 512 units and using the ReLU activation function, of the classification task. We use mini-batch SGD with batch size 256 to train this network.

Figure 5, Figure 6 and right panel of Figure 4 are based on the above setting.

---

[3]The curves for the three linear networks are almost identical and not visually distinguishable, we only present the one for 2-layer linear network in Figure 2.

**Multi-class classification on MNIST.** We use the following CNN to classify the 10 classes of MNIST. Specifically, this CNN contains two consecutive convolutional layers, with 32 and 64 channels, respectively. Both convolutional layers uses $3 \times 3$ kernel size and are with stride 1. On top of the convolutional layers, there is one max pooling layer, followed by two fully connected layers with width 64 and 10, respectively.

We injected $30\%$ random label noise into each class of MNIST training set. We use mini-batch SGD with batch size 512 to training the neural network.

Top row of Figure 7, left panel of Figure 8, and Figure 9 are based on this setting.

**Multi-class classification on CIFAR-10.** For the CIFAR-10 dataset, we use a standard 9-layer ResNet (ResNet-9) to classify [4]. We injected $40\%$ random label noise into each class of CIFAR-10 training set. We use mini-batch SGD with batch size 512 to training the neural network.

Bottom row of Figure 7 and right panel of Figure 8 are based on this setting.

## C ADDITIONAL EXPERIMENTAL RESULTS

### C.1 ANGLE $\theta_g$ DISTRIBUTION FOR MULTI-CLASS CLASSIFICATION

We experimentally verify the directional distributions of single-logit sample-wise gradients on MNIST dataset. We use the same CNN as in Figure 7, and evaluate the angle $\theta_g$ distributions at the network initialization. Specifically, given $c \in \{0, 1, \cdots, 9\}$, we consider the $c$-th output logit. Note that, according to the one-hot encoding, only class $c$ has label 1 and all the rest classes have label 0 on this logit. Hence, the binary classifier at logit $c$ is essentially a one-versus-rest classifier. For each of these binary classifiers, we look at the angle $\theta_g$ distributions of the corresponding single-logit sample-wise gradients.

As shown in Figure 9, each sub-plot corresponds to one logit. We can see that, for each $c$:

- The clean segment of class $c$ (green) has its single-logit sample-wise gradients concentrated at small angles $\theta_g$.
- The noisy segment of class $c$ (red) has its single-logit sample-wise gradients in the opposite direction of clean ones, concentrating at large angles and being symmetric to the clean segment. The noisy segment gradient $g_{noisy}^{(c)}(\mathbf{w}_0)$ (red dash line) is sharply opposite to $g_{clean}^{(c)}(\mathbf{w}_0)$, with $\theta_g$ almost $180°$.
- The distribution of "other" segment (blue), which contains the clean samples of all other classes, is clearly separated from the class $c$ distributions. Moreover, the component of segment gradient $g_{other}^{(c)}(\mathbf{w}_0)$ that is orthogonal to $g_{clean}^{(c)}(\mathbf{w}_0)$ clearly has non-trivial magnitude (as the $\sin \theta_g \sim \Theta(1)$).

All the above observation are align with our analysis for binary classifiers in Section 3 (compare with Figure 4 for example).

### C.2 SEGMENT LOSS DYNAMICS

Here, we show the dynamics for the segment losses, i.e., clean segment loss $L(\mathbf{w}; \mathcal{D}_{clean})$ and noisy segment loss $L(\mathbf{w}; \mathcal{D}_{noisy})$. Figure 10 shows the curves of these segment losses under different experimental settings: binary classification (same setting as in Figure 5); multi-classification for MNIST dataset (same setting as in top row of Figure 7); and multi-classification for CIFAR-10 dataset (same setting as in bottom row of Figure 7).

Obviously, under each experimental setting, the noisy segment loss $L(\mathbf{w}; \mathcal{D}_{noisy})$ becomes worse (increases) in the early stage and decreases in the later stage, which is align with the clean-priority learning dynamics.

---

[4]For the detailed architecture, we use the implementation in `https://github.com/cbenitez81/Resnet9/blob/main/model_rn.py`.

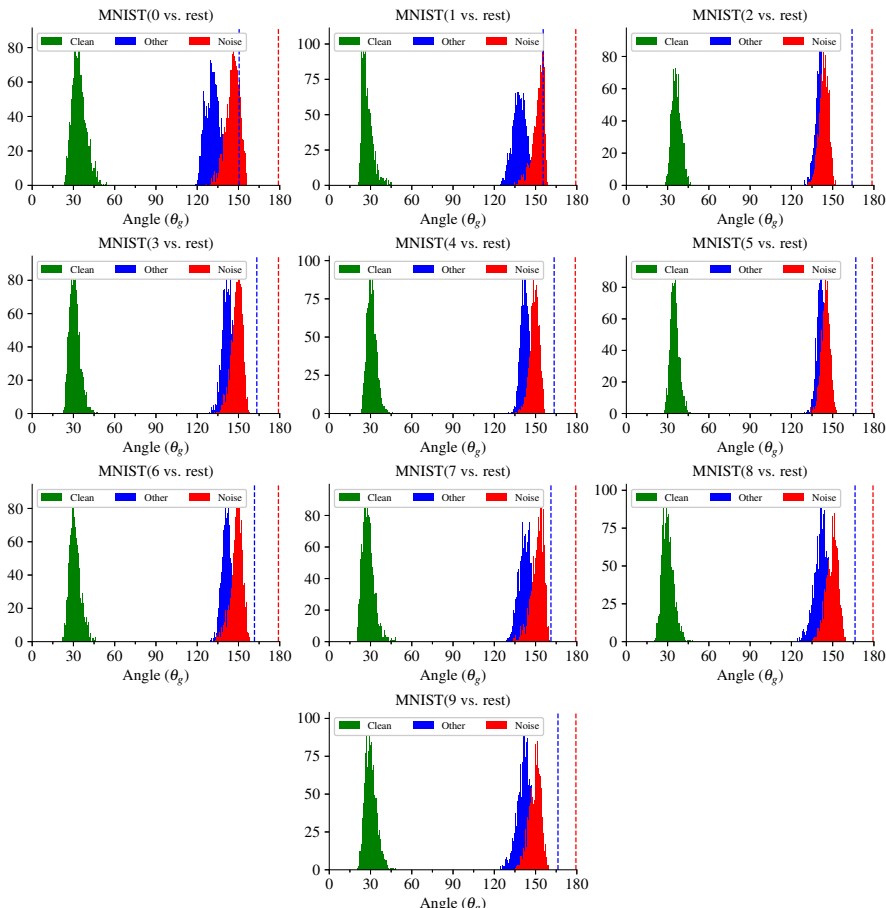

Figure 9: The distributions of $\theta_g$ for single-logit sample-wise gradients. MNIST dataset (label noise $\delta = 0.3$) on CNN. Dash lines represent segment gradients.

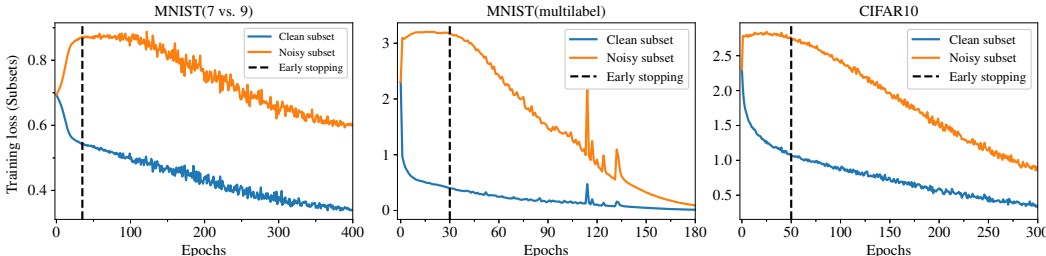

Figure 10: Training losses on clean and noisy segments. **Left**: for binary classification on two class MNIST ("7" and "9", noise level $\delta = 0.4$). **Middle**: for multi-class classification on MNIST (noise level $\delta = 0.3$). **Right**: for multi-class classification on CIFAR-10 (noise level $\delta = 0.4$).

### C.3 EFFECT OF WIDTH

We extract two classes, the images with digits "7" and "9", out from the MNIST datasets, and injected $40\%$ random label noise into each class in the training dataset (i.e., labels of $40\%$ randomly selected samples are flipped to the other class), leaving test set intact. We employ a fully connected neural network with 2 hidden layers. We sweep the number of neuran per layer from 32 to 2048 with ReLU activation function, of the classification task. We use mini-batch SGD with batch size 256 to train this network. It can be see in Figure 11 that the clean-priority learning is consistent with all widths.

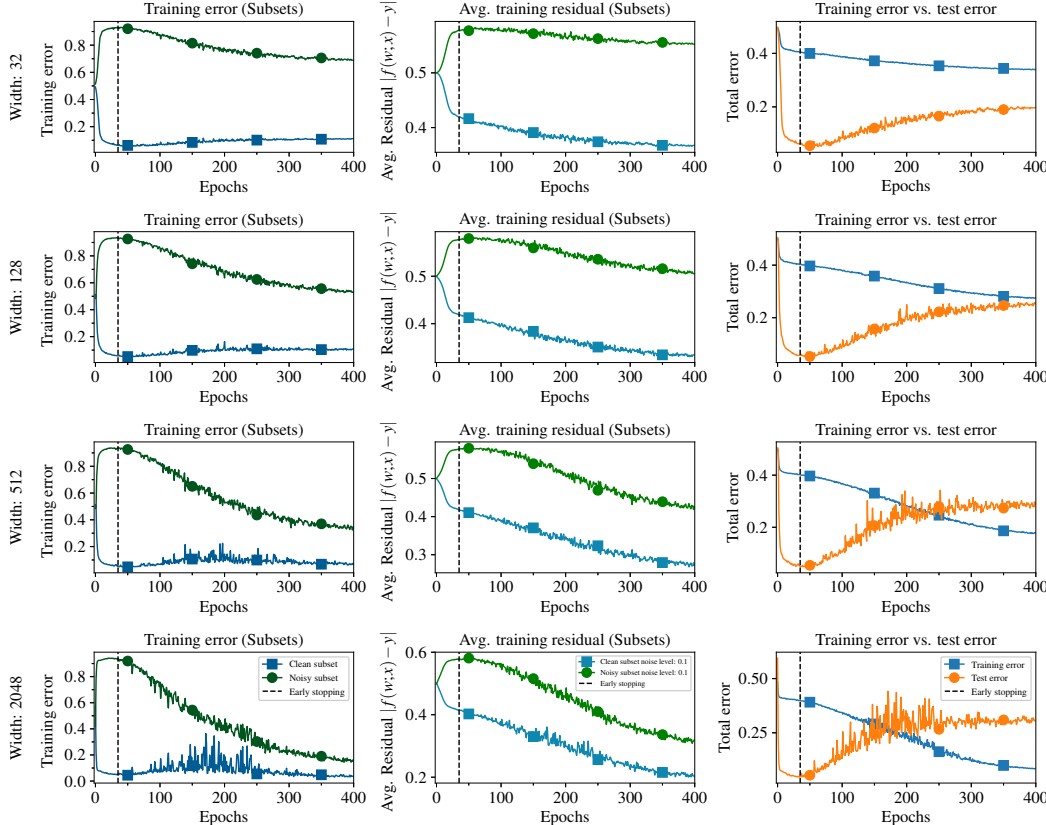

Figure 11: Learning dynamics on two classes ("7" and "9") of MNIST (noise level $\delta = 0.4$) with FC networks with different widths. **Left**: in the early stage (before the vertical dash line), clean segment error decreases, while noisy segment error increases. **Middle**: In the early stage, the clean segment average residual $\mathbb{E}_{(\mathbf{x},y) \in \mathcal{D}_{clean}}[|f(\mathbf{w}; \mathbf{x}) - y|]$ decreases, i.e., on average the network outputs of clean segment move towards the labels, indicating a "learning" on the clean segment. One the other hand, the noisy segment average residual, $\mathbb{E}_{(\mathbf{x},y) \in \mathcal{D}_{noise}}[|f(\mathbf{w}; \mathbf{x}) - y|]$, monotonically increases, indicating that the noisy segment is not-learned. **Right**: total test error and total training error.

## C.4   EFFECT OF NOISE LEVEL

We use the following CNN to classify the 10 classes of MNIST. Specifically, this CNN contains two consecutive convolutional layers, with 32 and 64 channels, respectively. Both convolutional layers uses $3 \times 3$ kernel size and are with stride 1. On top of the convolutional layers, there is one max pooling layer, followed by two fully connected layers with width 64 and 10, respectively.

We injected different level of random label noise from 0.1 to 0.4 into each class of MNIST training set. We use mini-batch SGD with batch size 512 to training the neural network. Figure 12 shows that clean priority is consistent for all noise levels.

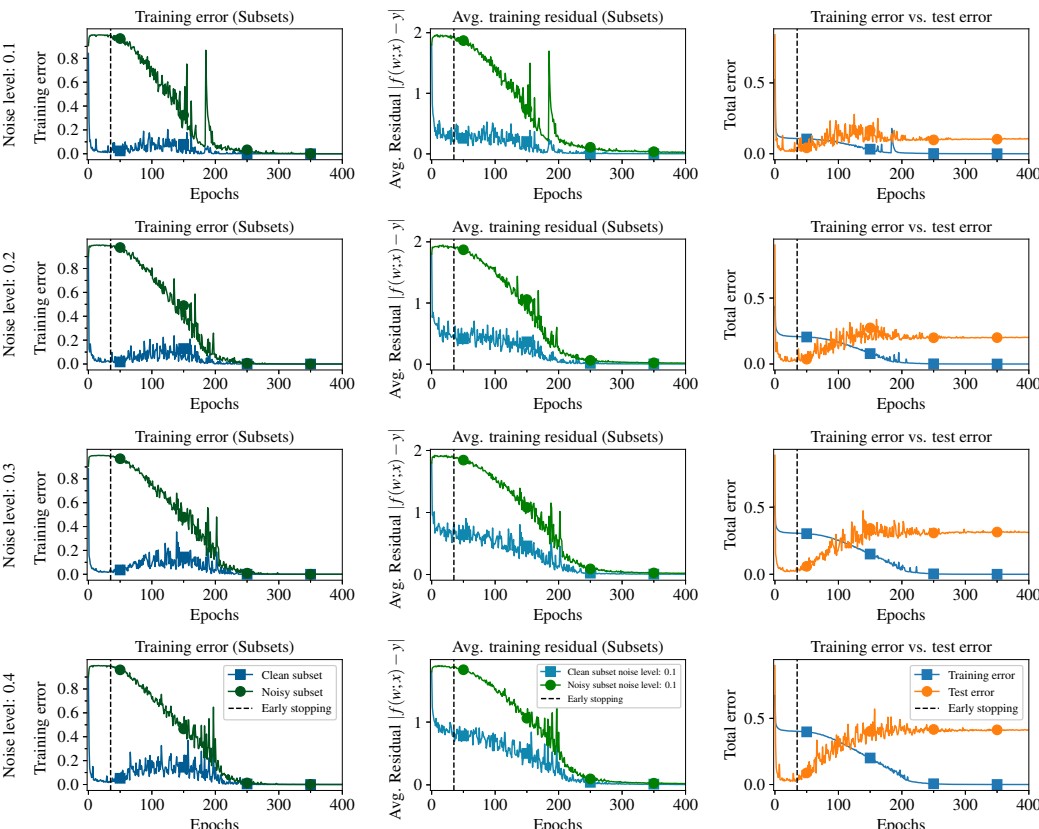

Figure 12: Learning dynamics on multi-class classification. **Left**: in the early stage (before the vertical dash line), clean segment error decreases, while noisy segment error increases. **Middle**: In the early stage, the clean segment average residual $\mathbb{E}_{(\mathbf{x},y)\in\mathcal{D}_{clean}}[\|f(\mathbf{w};\mathbf{x})-y\|]$ decreases, i.e., on average the network outputs of clean segment move towards the labels, indicating a "learning" on the clean segment. One the other hand, the noisy segment average residual, $\mathbb{E}_{(\mathbf{x},y)\in\mathcal{D}_{noise}}[\|f(\mathbf{w};\mathbf{x})-y\|]$, monotonically increases, indicating that the noisy segment is not-learned. **Right**: total test error and total training error.