# OpenReview forum: "Mechanism of clean-priority learning in early stopped neural networks of infinite width"
_ICLR.cc/2024/Conference — Submitted to ICLR 2024_

### Official Review · Reviewer_ofvF · 2023-10-28

**Soundness:** 2 fair
**Presentation:** 2 fair
**Contribution:** 1 poor
**Rating:** 3
**Confidence:** 4

**Summary:**

This paper studies the U-shaped learning behavior of neural networks when trained on datasets with label noise. Through an analysis of neural networks with infinite width, the authors demonstrate that these networks initially prioritize learning clean data due to the dominance of the gradient signal from clean samples over noisy ones. Subsequently, once the clean samples are adequately learned, the neural networks proceed to "memorize" the noisy data.

**Strengths:**

Understanding the U-shaped learning behavior is an important topic, and investigating gradient dynamics holds promise for unraveling the underlying mechanism.

**Weaknesses:**

One notable weakness of this paper is the omission of several relevant works. The U-shaped behavior under consideration has been investigated since at least 2017, if not earlier. Having only a one-page reference list is not enough. As this paper mainly focuses on the theoretical explanation, below I will highlight some theoretical contributions in this domain:

[1] Sanjeev Arora, et al. "Fine-grained analysis of optimization and generalization for overparameterized two-layer neural networks." ICML 2019.

Consider Question 1 in [1], *Why do true labels give faster convergence rate than random labels for gradient descent?* In comparison with the question asked in the second paragraph of your introduction, *how can the model tell clean samples from noisy ones ...*. At least to me, [1] and you study a similar question.

Applying the findings from [1] to respond to the question asked in this paper: neural networks trained with GD learn clean samples faster or in the early phase because true labels align well with top eigenvectors of the NTK matrix whereas projections of random labels are close to being uniform. While [1] studies a two-layer ReLU network, their analysis does not require the infinite width (although it should still be sufficiently wide).


[2] Madhu S. Advani, et al. "High-dimensional dynamics of generalization error in neural networks." Neural Networks 2020.

[3] Zhenyu Liao and Romain Couillet. "The dynamics of learning: A random matrix approach." ICML 2018.

[2,3] use Random Matrix Theory to study the over-training and early-stopping. They also provide insights into why models tend to learn clean data first and later overfit to noisy data. Notably, the analysis in [2] shares fundamental similarities with that in [1]. While [3] focuses solely on linear models, it's worth noting that, considering this paper allows the width of neural networks to go to infinity, the analysis presented here may not have a significant advantage.

Also see:

[4] Sheng Liu, et al. "Early-learning regularization prevents memorization of noisy labels." NeurIPS 2020.

[5] Yu Feng and Yuhai Tu. "Phases of learning dynamics in artificial neural networks in the absence or presence of mislabeled data." Machine Learning: Science and Technology 2021.

[6] Zitong Yang, et al. "Rethinking bias-variance trade-off for generalization of neural networks." ICML 2020.

They all attempt to explain the U-shaped behavior, and [4] provides a similar explanation for why gradients from clean data vanish with this paper. I will refrain from listing more, but additional relevant works do exist.

**Questions:**

1. It seems the dominance of clean samples gradient is due to the fact that the majority of training samples are not corrupted. Could you conduct experiments on $\delta>0.5$ (e.g., $\delta=0.8$), and track the gradients and the errors on clean and noisy data? Do the theoretical findings suggest that the neural networks will fit noisy samples first if $\delta>0.5$?

2. Can the analysis in this paper address epoch-wise double descent? Specifically, considering Figure 9 in the deep double descent paper (Nakkiran et al., 2021), given that this paper assumes neural network width goes to infinity (closer to the "Large Model" setting in Figure 9, Left), the testing curve should exhibit a "double descent" behavior rather than a U-shaped curve, according to the previous observation. Could you elaborate on this?

3. A minor point: The paper's title includes "early stopped" and "infinite width", yet the abstract mentions neither. Moreover, "early stopped" might not be a critical keyword; perhaps "early phase" or "early stage" would be more appropriate?

---

### Official Review · Reviewer_5A6o · 2023-10-31

**Soundness:** 2 fair
**Presentation:** 2 fair
**Contribution:** 2 fair
**Rating:** 3
**Confidence:** 4

**Summary:**

The paper deals with the mechanism of clean-priority learning in early stopped neural networks of infinite width. The authors investigate the behavior of neural networks when trained with datasets containing random label noise. They observe that the prediction error of a neural network on a noise-free test dataset initially improves during early training but eventually deteriorates, showing a U-shaped dependence on training time. The authors refer to this observed behavior as "clean-priority learning", where neural networks seem to learn the pattern of clean data first and fit the noise later. The study aims to explore the learning dynamics underlying this phenomenon. The authors demonstrate that in the early stages of training, the update direction of gradient descent is primarily determined by the clean samples of training data, minimizing the impact of noisy samples.

**Strengths:**

* Understanding the clean-priority learning mechanism holds significance in enhancing the robustness and generalization of neural networks, especially in the presence of noisy labels in the training data.

* This paper is structured in a clear and coherent manner.

**Weaknesses:**

* The paper’s reliance on Assumption 4.1 appears to be a strong precondition.

* The theoretical framework for infinitely-wide neural network utilized in the study seems to lack novelty. Since it's known as the lazy training [1], and can't learn feature from the data [2].

*  The presentation of the theoretical content in the paper seems to lack a coherent and logical flow. The sections appear somewhat disconnected, where later parts of the theory do not rigously build upon or connect with the earlier parts.

* The theoretical characterization presented in the paper appears to be somewhat overly detailed, possibly making it challenging for readers to extract the key points and contributions. For example, in Theorem 4.4, this is only a relation between current step and next step.


[1] Chizat, Lenaic, Edouard Oyallon, and Francis Bach. "On lazy training in differentiable programming." Advances in neural information processing systems 32 (2019).

[2] Damian, Alexandru, Jason Lee, and Mahdi Soltanolkotabi. "Neural networks can learn representations with gradient descent." Conference on Learning Theory. PMLR, 2022.

**Questions:**

* Could you elaborate on the choice of the theoretical framework, specifically focusing on infinitely-wide neural networks and lazy training? How does this choice enhance the novelty and significance of the study in the context of existing literature?

* Could you provide a more thorough comparison with other related works focusing on explaining the dynamics of label noise in neural network training? How does your approach and findings differentiate from or advance beyond existing explanations and theories in this area?

* Are there any experiments conducted specifically on infinitely-wide neural networks in the study?

---

### Official Review · Reviewer_aNMp · 2023-10-31

**Soundness:** 3 good
**Presentation:** 3 good
**Contribution:** 2 fair
**Rating:** 3
**Confidence:** 3

**Summary:**

The authors seek to explain the learning dynamics present when there is random label noise. They argue that the noisy and clean gradients are in opposite directions, and the later stage of learning with memorization occurs when the ratio of the norms between the clean and noisy gradients approaches 1. This analysis is then extended from the binary classification setting to the multi-class classification setting.

**Strengths:**

The authors provide a theoretically grounded explanation for an empirical phenomenon. The main results are clearly presented.

**Weaknesses:**

The assumptions on the data are very strong, and stand in stark contrast to the statements in the "Related Works" section. "Namely, each noisy sample must be covered by more clean samples. This assumption is often not met by actual datasets." It seems to me that the assumptions on the dataset here are much more restrictive, in order to satisfy Assumption 4.1.

There is no theoretical analysis of what happens in the later stage.

**Questions:**

Is my assessment that Assumption 4.1 imposes strong assumptions on the dataset correct? Is Assumption 4.1 satisfied for the MNIST experiments conducted?

How do the assumptions here compare to the dataset assumptions in prior work?

---

### Official Review · Reviewer_dh81 · 2023-11-01

**Soundness:** 3 good
**Presentation:** 3 good
**Contribution:** 3 good
**Rating:** 5
**Confidence:** 2

**Summary:**

The paper investigates the U-shaped dependence of neural network prediction errors on training time when random label noise is introduced to the training dataset. This U-shaped behavior is attributed to "clean-priority learning," where neural networks prioritize learning from clean data before fitting to the noise. The study reveals that during early training, gradient descent updates are primarily influenced by clean samples, rendering noisy samples inconsequential. However, as training progresses, the influence of clean samples on gradient updates wanes, leading to the eventual fitting of noisy samples. The paper's key contributions include: 1) Identifying that noisy samples' impact on gradient descent updates is negated by clean samples in early training, 2) Experimentally confirming that the dominance of clean samples decreases over time, especially around the early stopping point, and 3) Providing a theoretical proof for these observations in the context of fully connected networks.

**Strengths:**

Strength:
1. Paper is well organized and motivated.
2. The observation in Figure 4 about the angles between clean and noise labels are interesting and leads to reasonable assumptions.

**Weaknesses:**

Weakness:
I think this work contributes a bit marginally.
1. This “U”-shape of test error and the root in the dominance of the gradients from clean data in early training, are explained in previous work, see [1]. Therefore, this work does not substantially advance our knowledge of learning dynamics of noise labels.
2. I don’t see significant contributions from the theoretical techniques. It is still based on the analysis of infinitely wide networks at initialization. Please kindly clarify the technical contributions on the theory part.

[1] “Early-learning regularization prevents memorization of noisy labels” 2020

**Questions:**

Question:
1. Eq. 5: I could not see why is this ratio in magnitudes? Is this also confirmed in experiments, for both early and late training stage, and both binary and multi-class cases?
2. Can we also confirm Figure 4 in multi-class setting?

---

### Meta-Review · Area_Chair_ozPc · 2023-12-09

**Metareview:**

This work studies the label noise robustness of neural networks. The idea is that network first prioritizes the clean labels and fits to label noise much later which results in a U-shaped test error behavior. While reviewers appreciate the theoretical contributions, they have found that the results and observations are rather incremental with respect to prior work.

**Justification For Why Not Higher Score:**

N/A

**Justification For Why Not Lower Score:**

N/A

---

### Decision · Program_Chairs · 2024-01-16

Reject